# In Silico Analysis of Nanoplastics’ and β-amyloid Fibrils’ Interactions

**DOI:** 10.3390/molecules28010388

**Published:** 2023-01-02

**Authors:** Silvia Gabbrielli, Luca Colnaghi, Gemma Mazzuoli-Weber, Alberto Cesare Luigi Redaelli, Alfonso Gautieri

**Affiliations:** 1Biomolecular Engineering Lab, Dipartimento di Elettronica, Informazione e Bioingegneria, Politecnico di Milano, Piazza Leonardo da Vinci 32, 20133 Milano, Italy; 2Division of Neuroscience, IRCCS San Raffaele Scientific Institute, Via Olgettina 60, 20132 Milano, Italy; 3School of Medicine, Vita-Salute San Raffaele University, Via Olgettina 58, 20132 Milano, Italy; 4Center for Systems Neuroscience (ZSN), 30559 Hannover, Germany; 5Institute for Physiology and Cell Biology, University of Veterinary Medicine Hannover, Foundation, 30173 Hannover, Germany

**Keywords:** nanoplastics, amyloids, protein aggregation, molecular dynamics simulations, coarse-grained models

## Abstract

Plastic pollution has become a global environmental threat, which leads to an increasing concern over the consequences of plastic exposition on global health. Plastic nanoparticles have been shown to influence the folding of proteins and influence the formation of aberrant amyloid proteins, therefore potentially triggering the development of systemic and local amyloidosis. This work aims to study the interaction between nanoplastics and β-amyloid fibrils to better understand the potential role of nanoplastics in the outbreak of neurodegenerative disorders. Using microsecond-long coarse-grained molecular dynamics simulations, we investigated the interactions between neutral and charged nanoparticles made of the most common plastic materials (i.e., polyethylene, polypropylene, and polystyrene) and β-amyloid fibrils. We observe that the occurrence of contacts, region of amyloid fibril involved, and specific amino acids mediating the interaction depend on the type and charge of the nanoparticles.

## 1. Introduction

There is growing concern about the consequences on human and animal health related to plastic exposition. Natural degradation of plastic results in the formation of microplastics (MPs) and nanoplastics (NPs). These are particles unintentionally produced with physical irregularity, polydisperse in size, and with a non-uniform surface depending on various environmental factors. There is no universally accepted definition of NPs, but most studies, according to a general definition of nanoparticles, define NPs as plastic particles with a diameter in the range of 1–100 nm [1,2].

Given the high level of plastic pollution and their small size, plastic nanoparticles can easily disperse into the atmosphere through the evaporation of water. Subsequently, precipitation spreads plastic into the environment, contaminating lakes, oceans, and other waterways [3]. The main routes of exposure to NPs for humans are ingestion (food chain, drinking water), skin exposure (water, cosmetic products), and inhalation (air pollution) [4,5]. After absorption, NPs can diffuse systemically and can accumulate in a variety of tissues and organs: the presence of NPs has been detected in lungs, blood, brain, and portions of the gastrointestinal system [5,6,7]. NPs are two orders of magnitude smaller than eukaryote cells; thus, they can also be internalized by cells [8] through different mechanisms depending on the particle size: smaller particles can enter the cell under passive diffusion and pinocytosis [9,10], and larger NPs can cross cell membranes via receptor-mediated pathways, such as clathrin-mediated endocytosis and caveolin-mediated endocytosis [10].

Once internalized, NPs can interact with biomolecules and cellular structures in the cytoplasm. Interaction between NPs and proteins can lead to three different consequences: protein corona formation, protein-induced coalescence of NPs, and conformational changes of protein secondary structure [11]. NPs immersed in biological fluids interact with proteins that surround the NPs to form a heterogeneous layer called the protein corona. Adsorption of biomolecules on the surface of NPs can provide nanoparticles a new biological identity: coronated NPs can overcome the protection mechanisms of the immune system, and they can interact with the receptors present on the cell membrane [12] and interfere in the molecular processes. The protein corona is composed of two layers: an inner layer, called the hard corona, made up of proteins that show a high binding affinity with the NP, and an external layer, or soft corona, which consists of proteins characterized by a low binding affinity with NP [2]. NP–protein interaction can also result in conformational modifications of the native structure of proteins. The function of a protein is closely related to its three-dimensional structure: potential changes in the structure can cause misfolding and loss of functionality. Hollóczki et al. [11,13] demonstrated, through molecular dynamics (MD) simulations, that interaction with NPs, such as polyethylene and polystyrene nanoparticles, can cause changes in the secondary structure of proteins.

NPs can cross the blood–brain barrier [14] and can interact with proteins and potentially cause conformational changes. This suggests that NPs can lead to the aggregation of misfolded proteins, potentially contributing to the evolution of neurodegenerative diseases [15]. Protein fibrillation is defined as a dynamic process during which misfolded proteins form large linear aggregates or amyloid fibrils. Several proteins and peptides can lead to the formation of amyloid fibrils, including β-amyloid (Aβ) peptides, prion proteins, α-synuclein, and β2-microglobulin proteins. Alzheimer’s disease (AD) is caused by the aggregation of β-amyloid fibrils, and the accumulation of Aβ structures is one of the first events that characterize the pathogenesis of this disease [16]. Aβ fibrils have a direct neurotoxic effect on cells but can also break down into smaller parts: oligomers made up of 25–35 peptides deriving from these fibrils are highly cytotoxic. The nucleation phase is thermodynamically unfavorable and represents the major limiting step that influences the speed of fibril formation. The process can be accelerated by preformed seeds [17] and, in this light, the presence of polymeric nanoparticle seeds can influence the rate of protein fibrillation [18]. An in vitro analysis by Linse et al. [19] demonstrated that the presence of nanoparticles increases the probability of the formation of nucleation seeds, and shorter nucleation times have been observed in the presence of NPs. These results suggest a nanoparticle-assisted fibrillization, in which the NPs act as nucleation sites. Cabaleiro-Lago et al. [20] found instead that the presence of polymeric nanoparticles causes a significant increase in activation time during the fibrillation of Aβ proteins. However, the interaction with the NPs does not completely block the formation of fibrils, and the elongation and saturation phases are unchanged. Another study by Cabaleiro-Lago et al. [21] investigates the dual effect of NPs on protein β-amyloids’ fibrillation: depending on the specific ratio between the peptide and particle concentration, the NP effects can vary from acceleration of the fibrillation process to inhibition.

While a growing body of work is indicating that NPs have the potential to influence amyloid formation [15], how NP features influence protein aggregation and the molecular mechanisms involved are still unclear. To provide insights into the molecular mechanism of NP–amyloid interactions, in this work we used molecular dynamics (MD) simulations to study the interaction between different Ps and Aβ42 fibrils. Similar approaches have been used in the past to investigate the interaction between proteins and nanoparticles or nanomaterials such as carbon nanotubes [22,23] and graphene [24,25]. Gold nanoparticles (AuNPs) are by far the most studied nanoparticles thanks to their relevance in biomedical applications. A seminal work by Brancolini et al. [4] studied the interaction between ubiquitin and AuNPs (modelled as a planar surface), characterizing the structure of the ubiquitin–gold surface complex, and providing insights into the driving forces for the protein–NP binding. An early work [5] studied the interaction between AuNPs of different shapes (size ≈ 12 nm) and albumin, the main protein in human blood plasma, showing that in the interaction with any shape of AuNPs, human serum albumin unfolds and that the content of alpha helix decreases. Conversely, a more recent work [6] showed that the albumin protein, despite a limited loss of secondary structure, remains in a folded structure and that the interaction with the AuNPs significantly decreases the flexibility of a large part of the protein. In a recent work, Tavanti et al. [7] modelled the interaction of relevant proteins (hemoglobin, myoglobin, and trypsin) with citrate-capped AuNPs of 15 nm diameter, determining the specific binding sites for each protein. By modelling the competition between proteins during the adsorption process, the authors were able to assess the final composition of the protein corona on the AuNPs. The authors did not observe any major conformational change in the protein structure because of the biding. Recently, Kalipillaia et al. [8] studied the interaction of amyloidogenic protein amyloid β (Aβ40) with AuNPs (3–5 nm in diameter) and characterized residue-specific contacts and the effects on the secondary structure of the protein, finding that AuNPs reduce the formation of beta sheets, thus preventing the formation of mature fibers.

In the present work, coarse-grained models of polystyrene (PS), polyethylene (PE), and polypropylene (PP) nanoparticles were developed, placed near a β-amyloid fibril, and then simulated for up to a microsecond. The simulations allowed us to characterize the occurrence of protein–plastic contacts and to identify the amino acids most involved in the interactions depending on the specific NPs.

## 2. Results and discussion

### 2.1. Nanoplastics Models

The NP models were generated starting from polymeric chains that self-assembled in short MD simulations due to weak hydrophobic residues. For all polymers (PS, PE, and PP), we obtained NPs with equal mass (80 kDa) and with an average diameter of 6 nm (Figure 1A–C). The NP molecular models lay in the lower range of NPs observed in the environment, but still represent realistic NPs of realistic size. The obtained nanoparticles are characterized by irregular shape and surface, which matches the characteristics of NPs that are found in the environment. Indeed, NPs derive from the fragmentation of larger plastic pieces, and for this reason they are characterized by irregular shapes. Since NP can present a net charge [26], we generated the model of charged polystyrene NP with different charge densities and signs. We were not able to generate reliable models of PS50+ (50% of positively charged monomers) and PS50- (50% of negatively charged monomers) since the electrostatic repulsion forces prevailed over attraction forces due to the high charge density. We obtained a reliable model for positive and negative NPs with charge densities of 4%, 10%, and 20%. As a result, charged NPs present a total absolute charge of ± 90*e* (p20 and n20) ± 45*e* (p10 and n10), and ± 18*e* (p4 and n4). At the end of the MD simulations, we observe that charges are distributed on the surface of the NPs, with only minimally charged beads located in the core of the NPs (Figure 1D).

### 2.2. Aβ Fibril Model

The investigation of the NP–fibril interactions focused on the characterization of the contacts between the two species. In the case of a contact, we identified a contact area and the interaction is classified as a *side contact* or a *frontal contact*. This can have different consequences on the fibrillation process. In frontal contact, almost exclusively hydrophobic residues are involved; in side contact, a mix of polar, charged, and hydrophobic residues are involved in the interaction (Figure 2A–B). Each Aβ monomer presents a negative charge -3*e*, since its sequence presents six negatively charged amino acids (D1, E3, D7, E11, E22, and D23) and three positively charged amino acids (R5, K16, and K28). Consequently, the fibril presents an overall negative charge. Given the position of the charged amino acids in the fibril structure, it is possible to identify two highly charged areas, where electrostatic interactions with the NPs may occur. The first region is the corner loop made by the negatively charged amino acids E22 and D23. The second region is the face defined by residue D1 to E11 with the addition of K28 from a separate monomer. This region features a mix of positive and negative charges (Figure 2C).

### 2.3. NP–Aβ Fibril Interactions

We used MD simulations of systems comprising an Aβ fibril and an NP to characterize their interaction depending on the type and charge of the NP. Specifically, we characterized the number of total contacts, the number of hydrophobic contacts, and the number of polar/charged contacts. A deeper analysis involved the characterization of the specific amino acids of the Aβ fibril involved in the contact with the NP. The analysis of trajectories was carried out only in the cases where a contact was observed during the MD simulation.

The interaction between the protein fibril and the PS nanoparticles was observed in six out of eight simulations (Figure 3A). In most positive cases, the interaction took place very quickly (within the first 200 ns) and then the contact was maintained throughout the whole simulation. All contacts occur as side contacts, independently of the initial position of the NP. The amino acids mostly involved in the interaction are the hydrophobic residues placed on the side of the fibril. In particular, the contact is highly localized in correspondence with the fibril region that features aromatic amino acids (F19 and F20), in line with the chemical composition of PS, where the monomers present aromatic rings.

The systems with the PE nanoparticles lead to fibril–NP interaction in seven cases out of eight simulations (Figure 3B). The initial contact occurred between 50 and 800 ns, and it was maintained throughout the MD simulations. Concerning the type of contacts, no clear trends were observed since we detected a side contact in four cases and a frontal contact in three cases. The residues involved in the contact were mostly hydrophobic amino acids, although no clear pattern was observed as in the case of PS.

Furthermore, in the case of PP nanoparticles, we observed the contacts in seven cases out of eight simulations (Figure 3C). However, for these NPs, we observed five frontal contacts and two side contacts. The time required for the initial contact was between 50 ns and 500 ns. The interaction between PP nanoparticles and the Aβ fibril was mediated by hydrophobic interactions, although no clear evidence of a specific amino acid mediating the contact could be observed.

When analyzed collectively, the fibril–NP simulations provided insights into the amino acids mostly involved in the interaction. Independently of the initial position and the type of the plastic material, the amino acids 16-20 (K16, L17, V18, F19, and F20) were the most involved in the interaction with the NPs (Figure 4A).

To understand the role of charged NPs and their interaction with Aβ fibril, we generated PS nanoparticles with different charge signs and densities. Table 1 summarizes the observed contact type. We observed that the negatively charged NPs do not interact with the fibril over the course of the simulations, independently of the charge density. This behavior is expected due to the overall negative charge of the Aβ fibril, which leads to electrostatic repulsive forces overcoming the attractive forces due to hydrophobic interactions. On the other hand, we observed long-lived interactions between the Aβ fibril and the positively charged NPs (Figure 3D–E), with contact areas mostly localized on the side of fibril. It was observed that negatively charged amino acids of fibrils, such as E11 and E22-D23, are frequently involved in the interaction, in addition to neighboring F19 and F20. Therefore, the results suggest that positively charged PS nanoparticles interact with the fibril mostly through aromatic interactions and electrostatic interactions (Figure 4B).

We analyzed the changes in the structure of the fibril by assessing the content of the beta-sheet secondary structure (Figure 5A). Concerning the crystal structure, the MD simulations show a decrease in the beta-sheet content, due to the higher mobility of the fibril at room temperature and the partial unfolding of the terminal peptides. However, no significant changes in the beta-sheet content are observed due to the interaction with the different NPs. Concerning the NPs, we tested whether the interaction with the amyloid fibril leads to changes in the shape of the particles. We calculated a shape factor (*SF*)—based on the moments of inertia along the three orthogonal axes—which assumes values of 0 for a perfectly rounded particle and a value of 1 for a one-dimensional fibril. We observed the initial values of the *SF* in the range of 0.25–0.40, indicating a roughly spherical shape (Figure 5B). The interaction with the fibril consistently leads to a slight increase in the *SF*, inducing a more elongated structure, although no major changes were observed. These changes in the NPs structure were not observed for the negatively charged NPs, as a result of the lack of any significant interaction with the protein.

## 3. Conclusions

This work focused on the investigation of the interaction between plastic nanoparticles of PS, PE, PP, and β-amyloid fibrils. The results of coarse-grained MD simulations showed that NPs mainly interact with hydrophobic regions of the β-amyloid fibril, and specifically aromatic residues in the case of PS. Negatively charged NPs do not interact with the fibril, whereas positively charged NPs show significant contacts mediated by both electrostatic and aromatic interactions in the region F19-D23. Overall, the contacts do not lead to a disassembly or major unfolding of the fibril during the course of the simulations, suggesting that the presence of the NPs does not alter the structure of fibrils that are already formed.

The analysis of the type of contact indicates that the presence of NPs can potentially influence Aβ protein fibrillation. The majority of contacts for PS nanoparticles occur on the side of the fibril, which would not prevent the addition of new monomers during the aggregation process. Previous work has shown that the fibrillation process can be accelerated by preformed seeds [17,18]. In this light, NPs can act as nucleation sites and/or accelerate the fibrillation process by stabilizing Aβ oligomers via side interaction, without blocking fibril elongation and, thus, enhancing the probability to form mature amyloid fibrils. In the case of PE and PP nanoparticles, we observed frontal contact with the amyloid fibril. Stable interactions on the frontal area of the fibril may inhibit the progress of fibrillation because the presence of NP on the fibril surface would block the addition of new monomers. Therefore, PE and particularly PP types of NPs could potentially delay the fibrillation process and increase the content of the more cytotoxic oligomers.

Our results contribute to understanding how NPs in the environment interact with protein and may pose a health issue. The models developed in this work are of necessity a simplified description presenting some limitations. The NPs models present a small size, and this limitation is mainly due to the computational cost, which scales as ≈N^2^ (where N is the number of beads in the system). Modelling an NP with a diameter 10 times larger (e.g., 60 nm) would lead to a particle volume 1000× larger and a computational cost of approximately 1 million times larger. For this reason, the current computational resources limited the NP size to a few nm in diameter in this study, as reported in other similar works’ diameters [1,2,3]. Nonetheless, the models represent realistic NPs although at the lower end of the environmentally relevant range, and the constantly increasing computational power will allow for a scale-up in the size of modeled NPs. In addition, NPs are known to quickly interact with proteins and other biomolecules and to be surrounded by a protein corona, which is not considered in this work. However, the protein corona may significantly affect the behavior and biological interactions of the NPs. Understanding the composition and structure of the protein corona and its influence on the biological effects of NPs is one of the key future challenges in the field.

## 4. Materials and Methods

All MD simulations are implemented using GROMACS simulation software [27] in combination with the MARTINI force field [28] optimized for coarse-grained modelling of biomolecules. The CG MARTINI force field uses a “4-to-1” mapping model in which four heavy atoms and their hydrogens are represented by a single interaction site, or *bead*. In this way, it is possible to reduce the number of particles and simulate larger systems and longer times. The MARTINI force field provides parameters for a large number of biochemical building blocks, and it allows us to build topologies of a wide range of biomolecules, including proteins, lipids, and nucleic acids [29]. In addition, the MARTINI force field can be extended to other molecules, such as polymers. The systems considered in this work are made of two main elements: the CG model of fibril Aβ (1-42) and the CG model of an NP (either polystyrene (PS), polyethylene (PE), or polypropylene (PP)).

### 4.1. Aβ Fibril Model

The CG model of the amyloid fibril was obtained starting from Aβ 1-42 fibrillary structure, available on the Protein Data Bank (PDB ID: 5OQV [30]). This structure is formed by nine identical chains of 42 amino acids, with a high content of beta-sheet structures. The 5OQV structure was modified using VMD [31], eliminating one of the nine chains, to obtain a basic unit with a symmetrical structure. This block was replicated and translated four times along the longitudinal axis, to obtain a single Aβ oligomer made of 32 protein chains and ≈8 nm in length. This structure was minimized and equilibrated in water, at constant pressure (1 atm) and temperature (300 K). The atomistic structure of Aβ fibril (Figure 6A) was then converted into a coarse-grained structure (Figure 6B).

### 4.2. Nanoplastics Models

The coarse-grained NP models of PP, PE, and PS were generated using parameters available in the literature [32,33]. To build the nanoparticle models, we initially constructed single polymeric chain made of 50–160 monomers (depending on the material), then generated a system with multiple copies (9–18) of the linear chains, and finally equilibrated the systems for 100 ns in vacuum to obtain spheroidal nanoparticles of ≈80 kDa and ≈6 nm in diameter for each plastic material (Figure 7). The molecular models of the NPs (coordinates and topologies) can be found in Appendix A. To investigate NP–fibril interactions in the case of charged NPs, we build also a set of six charged polystyrene NPs with negative and positive charge and different charge densities (named PS4+, PS10+, PS20+, PS50+, PS4−, PS10−, PS20−, and PS50−, where +/− indicate the positive or negative charge, and the number is the percentage of charged beads in the NP).

### 4.3. MD Simulations of Fibril–NP Complexes

For each NP, were built eight distinct systems consisting of a fibril and an NP using eight different random initial configurations (Figure 8) to avoid bias due to the initial orientation of the NP concerning the fibril. Independently on the orientation, the NP was placed with an initial spacing > 4 nm. A total of 80 different systems were built (8 replicas × 9 NP species, and 8 replicas of the fibril without NPs).

The systems were placed in the center of a 20 mm × 20 mm × 20 nm periodic box, allowing for a distance of at least 2 nm between solute and the boundaries of the box. The systems were then solvated using the MARTINI CG water model, and counter ions were introduced to keep the periodic systems neutral. The models resulted in boxes with ≈70,000 beads and 20 nm in length. Following energy minimization, all the systems were simulated for 1000 ns at constant pressure (1 atm) and temperature (300 K) using a time step of 20 fs.

To study the interaction between fibrils and NPs, different types of analysis were performed, focused on the characterization of the contacts between the NP and the amino acids of the fibril. In particular, we analyzed the total number of contacts (i.e., the number of fibril amino acids within a distance cut-off of 8Å from the NP), the contacts made from a different class of amino acids (hydrophobic, polar, charged), and the specific residues most involved in the interactions. For the analysis of the fibril secondary structure content, the CG model at the end of the simulations was backmapped to an atomistic model, and a 1 ns equilibration was performed before the secondary structure analysis. To assess the shape of the NPs, we defined a “shape factor” (*SF*) parameter. First, the moments of inertia (MOIs) along the principal axes of the NPs are calculated using the *gyrate* tool of GROMACS. Then, the NPs are aligned along the principal axis imposing that the NP dimension with the lowest MOI is aligned along the x-axis. The *SF* term is then calculated as:

$$SF=1-\frac{I_{x}}{\left(I_{y}+I_{z}\right)/2}$$
where $I_{x},\;I_{y},\;\mathrm{and}\;I_{z}$ are the MOI along the *x*-, *y*-, and *z*-axis, respectively. An *SF* close to 1 identifies an elongated, fibril-like NP, whereas an *SF* close to 0 indicates a rounded NP. All analyses were performed in VMD using in-house scripts [34,35,36].

## Figures and Tables

**Figure 1 molecules-28-00388-f001:**
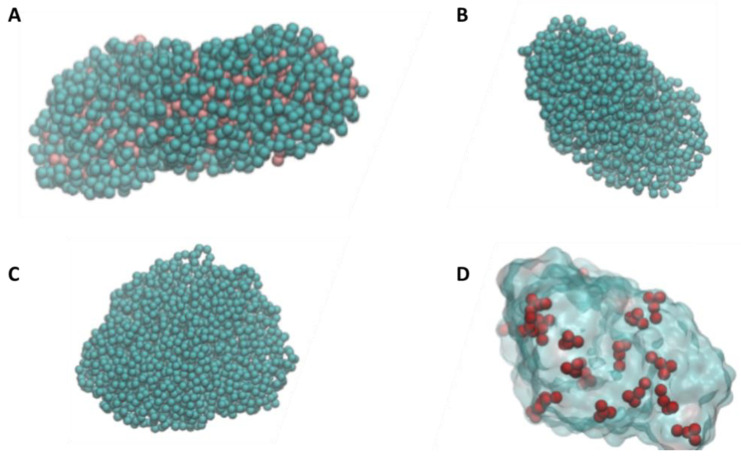
Representative molecular models of the different NP. (**A**) Polystyrene, where pink beads represent the (CH_2_-CH) groups, while cyan beads represent the phenyl groups; (**B**) Polyethylene; (**C**) Polypropylene; and (**D**) Polystyrene with 10% of positively charged monomers (positively charged monomers are represented in red color).

**Figure 2 molecules-28-00388-f002:**
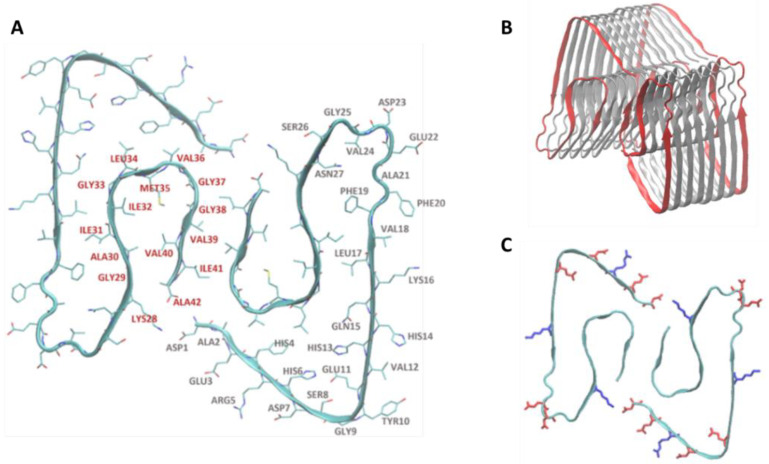
Molecular model of the Aβ fibril. (**A**) Frontal view of Aβ (1-42) fibrillary structure: in red are represented residues of group A, and in gray the residues of group B. (**B**) Side contact (gray) and frontal contact (red) areas. (**C**) Representation of two “1–42” amino acid chains that compose the fibril. The positively charged residues are highlighted in blue, while the negatively charged residues are highlighted in red.

**Figure 3 molecules-28-00388-f003:**
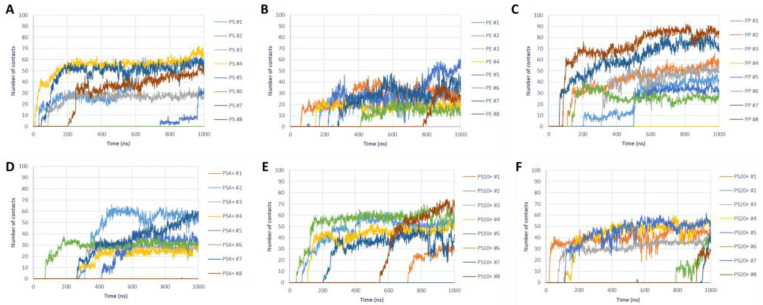
Number of contacts as a function of time. The plots show the number of amino acids in contact with the NPs for nanoparticles made of (**A**) PS; (**B**) PE; (**C**) PP; and positively charged PS nanoparticles with a charge density of (**D**) 4%, (**E**) 10%, and (**F**) 20%. For the negatively charged NPs, we did not observe interactions during the MD simulations.

**Figure 4 molecules-28-00388-f004:**
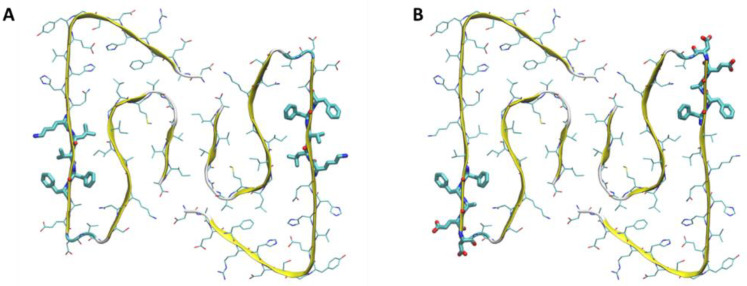
Amino acids most involved in the interactions (**A**) with the neutrally charged NPs and (**B**) with the positively charged NPs.

**Figure 5 molecules-28-00388-f005:**
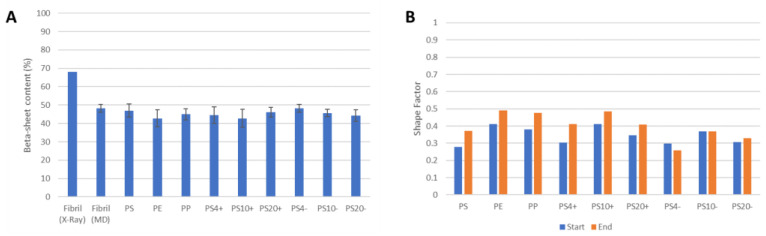
Structural effects of the protein–NP interaction. (**A**) Beta-sheet content of the fibril structure, showing that the interaction with the NPs does not significantly alter the beta-sheet content of the fibril. The shape factors of the NPs (**B**) at the start and the end of the simulations show that the interaction with the amyloid fibril does not lead to major changes in the shape of the NPs, although the NPs assume a more elongated structure, as shown by the increasing *SF*.

**Figure 6 molecules-28-00388-f006:**
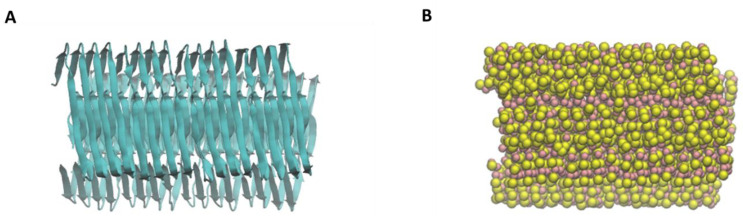
Molecular model of the amyloid fibril. (**A**) Visualization of atomistic structure of the fibril. (**B**) Visualization of coarse-grained structure of the fibril: pink beads correspond to the backbone of protein structure.

**Figure 7 molecules-28-00388-f007:**
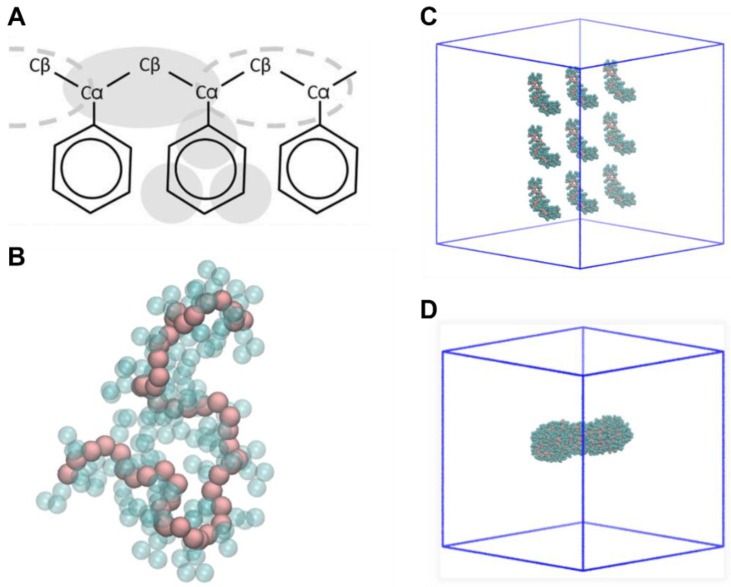
Schematics of the NP molecular models’ construction. (**A**) CG mapping of polystyrene [33]. (**B**) PS50: single linear chain, made up of 50 monomers of styrene. (**C**) Initial configuration: nine PS50 chains. (**D**) Final model of PS nanoplastic.

**Figure 8 molecules-28-00388-f008:**
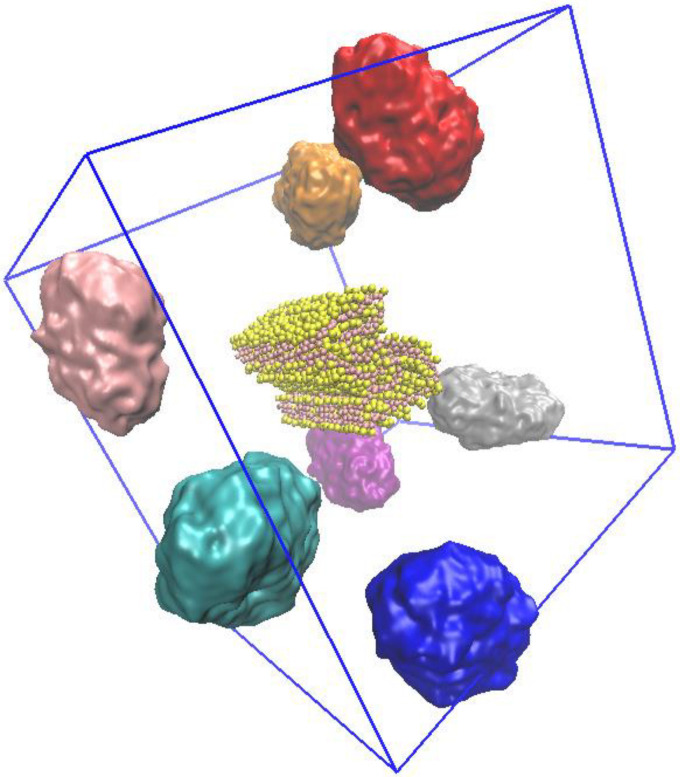
Fibril–NP molecular models. Overlap of the eight initial configurations for PE models, showing the protein fibril in the center and the initial positions of the NPs. Water and ions are not shown for clarity.

**Table 1 molecules-28-00388-t001:** Type of contact for the different NP–fibril systems.

	PS	PE	PP	PS4+	PS10+	PS20+	PS4-	PS10-	PS20-
System 1	side	frontal	side	-	side	side	-	-	-
System 2	-	side	frontal	frontal	frontal	-	-	-	-
System 3	side	-	frontal	side	-	side	-	-	-
System 4	side	side	-	side	side	side	-	-	-
System 5	side	frontal	frontal	side	side	side	-	-	-
System 6	-	side	frontal	side	-	side	-	-	-
System 7	side	side	side	frontal	side	side	-	-	-
System 8	side	frontal	frontal	-	frontal	side	-	-	-

## Data Availability

Data are available from the authors.

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
