# Peer review of "In Silico Analysis of Nanoplastics’ and β-amyloid Fibrils’ Interactions"

_molecules, 2023, doi:10.3390/molecules28010388_

Round 1

Reviewer 1 Report

Manuscript „ In silico analysis of nanoplastics and β-amyloid fibrils interactions“ by Silvia Gabbrielli et. al. is MD analysis of interaction between nanoparticles and protein fibrils. Taking into account huge problem with pollutions via plastic particles it is study of great social demand. Authors performed calculation on few kind of plastic nanoparticles formed by  polyethylene, polypropylene and polystyrene. As protein aggregates the β-amyloid fibrils were selected. Authors have tried to get the occurrence of protein-plastic contacts and to identify the amino acids most involved in the interactions. Manuscript is good written and represent interest for broad scientific community. Of course, there are few points which should be improved:

1.    The impact of this study for real system is not clear and was not estimated. Authors have done calculation for NP of 6 nm in diameter. Please estimate how does  this interaction effect whole system of plastic particles and fibrils. Is any up scaling possible.

2.    Please compare results with previous modelling of other particles. Some results as interaction between appositively  charged NPs and fibril is trivial.

3.      It looks that occurrence of interaction (3 from 4) between NPs and fibril does not depend on kind nanoparticles.  Can it be extended for other materials (gold nanoparticles?).

Reviewer 2 Report

The authors used molecular dynamics methods to simulate the interactions between neutral and charged nanoparticles. This topic is very interesting and also very important for both environment protection and human health. The results are very valuable, they will help us to better understand the interaction mechanism between NP and human cells. The article is well organized. The authors might pay a lot of time to polish their paper. I don't have comments. Hence, I advise the editor to accept this paper for publication.

Reviewer 3 Report

The authors present a report about interactions between nanoplastic (NPs) and β-amyloid fibrils. Coarse-grained models of NPs and β-amyloid fibrils were built up and then submitted to microsecond long molecular dynamics simulations. The simulation results show that NPs mainly interacts with hydrophobic regions of the β-amyloid fibrils, especially the aromatic residues F19 and F20 in the case of polystyrene. Besides, positively charged polystyrene particles could maintain contact with the fibril through both electrostatic and aromatic interactions, while negatively charged polystyrene particles do not interact with the fibril. These simulation results can be potentially interesting and may provide insights for understanding the NP effects (acceleration or inhibition) on fibrillation process. However, considering the interpretation of simulation results are rather limited and unconvincing, the manuscript is not suitable for publication before major revisions.

- Although multiple microsecond long molecular dynamics simulations were performed, the analyses about simulation results were limited, mainly on protein-NP contact number and contact residues. Whether and how the conformation of β-amyloid fibril changed (for example, bend) during simulations? Whether and how the conformation of NP changed (for example, elongate) during simulations? Why the NPs always failed to interact with protein in at least one out of four cases (for example, distance too far)? More simulation results should be provided, especially to support the conclusion that “the contacts do not lead to a disassembly or major unfolding of the fibril during the course of the simulations” (line 212).

-In Figure 1, the difference between red (subfigures A, D) and green beads (subfigures A-C) is not clear. Are the red beads hydrophobic or in charge? Please include more detail in figure legend.

-Line 180. “…were the most involved in the interaction with the NPs (Figure 7A).” It seems Figure 6A, instead of Figure 7A, is related to this statement.

-In conclusion section, Line 216-227, “The analysis of the type of contact indicates that… increase the content of the more cytotoxic oligomers.” The conclusions in this paragraph based on just four simulations are unconvincing. As the authors pointed out “no clear trends were observed since we detected a side contact in two out of three cases and a frontal contact in one case” (line 170)”. Therefore, a solid conclusion cannot be made just because one more positive case, especially considering the NPs always failed to contact with protein in one of the four simulations (repeatability is low). The statements in this paragraph are more like discussions instead of conclusions.

-Line 271, “…the systems for 100 ns in vacuum”. Were the simulations with protein also performed in water box instead of in vacuum? More detail about simulation method should be included.

Round 2

Reviewer 3 Report

This paper describes the interactions between nanoplastics and β-amyloid fibrils based on microsecond-long coarse-grained molecular dynamics simulations. The paper has been appropriately modified to improve the statistical analysis results. Thus, I think that this paper is now appropriate for publishing in Molecules.